ps # PLOS GENETICS

# Simultaneous SNP selection and adjustment for population structure in high dimensional prediction models

Sahir R. Bhatnagar[1,2]*, Yi Yang[3], Tianyuan Lu[4,5], Erwin Schurr[6], JC Loredo-Osti[7], Marie Forest[8], Karim Oualkacha[9], Celia M. T. Greenwood[1,4,5,10,11]

1 Department of Epidemiology, Biostatistics and Occupational Health, McGill University, Montréal, Québec, Canada, 2 Department of Diagnostic Radiology, McGill University, Montréal, Québec, Canada, 3 Department of Mathematics and Statistics, McGill University, Montréal, Québec, Canada, 4 Quantitative Life Sciences, McGill University, Montréal, Québec, Canada, 5 Lady Davis Institute, Jewish General Hospital, Montréal, Québec, Canada, 6 Department of Medicine, McGill University, Montréal, Québec, Canada, 7 Department of Mathematics and Statistics, Memorial University, St. John's, Newfoundland and Labrador, Canada, 8 École de Technologie Supérieure, Montréal, Québec, Canada, 9 Département de Mathématiques, Université du Québec à Montréal, Montréal, Québec, Canada, 10 Gerald Bronfman Department of Oncology, McGill University, Montréal, Québec, Canada, 11 Department of Human Genetics, McGill University, Montréal, Québec, Canada

* sahir.bhatnagar@mcgill.ca

## Abstract

Complex traits are known to be influenced by a combination of environmental factors and rare and common genetic variants. However, detection of such multivariate associations can be compromised by low statistical power and confounding by population structure. Linear mixed effects models (LMM) can account for correlations due to relatedness but have not been applicable in high-dimensional (HD) settings where the number of fixed effect predictors greatly exceeds the number of samples. False positives or false negatives can result from two-stage approaches, where the residuals estimated from a null model adjusted for the subjects' relationship structure are subsequently used as the response in a standard penalized regression model. To overcome these challenges, we develop a general penalized LMM with a single random effect called `ggmix` for simultaneous SNP selection and adjustment for population structure in high dimensional prediction models. We develop a blockwise coordinate descent algorithm with automatic tuning parameter selection which is highly scalable, computationally efficient and has theoretical guarantees of convergence. Through simulations and three real data examples, we show that `ggmix` leads to more parsimonious models compared to the two-stage approach or principal component adjustment with better prediction accuracy. Our method performs well even in the presence of highly correlated markers, and when the causal SNPs are included in the kinship matrix. `ggmix` can be used to construct polygenic risk scores and select instrumental variables in Mendelian randomization studies. Our algorithms are available in an R package available on CRAN (https://cran.r-project.org/package=ggmix).

sahirbhatnagar/ggmix/tree/master/simulation.
GAW20 and UK Biobank data are available via
application directly to Genetic Analysis Workshop
(https://www.gaworkshop.org/data-sets) and UK
Biobank (https://www.ukbiobank.ac.uk/using-the-
resource/).

**Funding:** SRB and CMTG were supported by the
Ludmer Centre for Neuroinformatics and Mental
Health and the Canadian Institutes for Health
Research PJT 148620. CMTG was also partially
supported by NSERC RGPIN-2019-04482. This
research was enabled in part by support provided
by Calcul Québec (www.calculquebec.ca) and
Compute Canada (www.computecanada.ca). YY
was partially supported by NSERC RGPIN-2016-
05174 and FRQ-NT NC-205972. The funders had
no role in study design, data collection and
analysis, decision to publish, or preparation of the
manuscript.

**Competing interests:** The authors have declared
that no competing interests exist.

## Author summary

This work addresses a recurring challenge in the analysis and interpretation of genetic
association studies: which genetic variants can best predict and are independently associ-
ated with a given phenotype in the presence of population structure? Not controlling con-
founding due to geographic population structure, family and/or cryptic relatedness can
lead to spurious associations. Much of the existing research has therefore focused on
modeling the association between a phenotype and a single genetic variant in a linear
mixed model with a random effect. However, this univariate approach may miss true asso-
ciations due to the stringent significance thresholds required to reduce the number of
false positives and also ignores the correlations between markers. We propose an alterna-
tive method for fitting high-dimensional multivariable models, which selects SNPs that
are independently associated with the phenotype while also accounting for population
structure. We provide an efficient implementation of our algorithm and show through
simulation studies and real data examples that our method outperforms existing methods
in terms of prediction accuracy and controlling the false discovery rate.

## Introduction

Genome-wide association studies (GWAS) have become the standard method for analyzing
genetic datasets owing to their success in identifying thousands of genetic variants associated
with complex diseases (https://www.genome.gov/gwastudies/). Despite these impressive find-
ings, the discovered markers have only been able to explain a small proportion of the pheno-
typic variance; this is known as the missing heritability problem [1]. One plausible reason is
that there are many causal variants that each explain a small amount of variation with small
effect sizes [2]. Methods such as GWAS, which test each variant or single nucleotide polymor-
phism (SNP) independently, may miss these true associations due to the stringent significance
thresholds required to reduce the number of false positives [1]. Another major issue to over-
come is that of confounding due to geographic population structure, family and/or cryptic
relatedness which can lead to spurious associations [3]. For example, there may be subpopula-
tions within a study that differ with respect to their genotype frequencies at a particular locus
due to geographical location or their ancestry. This heterogeneity in genotype frequency can
cause correlations with other loci and consequently mimic the signal of association even
though there is no biological association [4, 5]. Studies that separate their sample by ethnicity
to address this confounding suffer from a loss in statistical power due to the drop in sample
size.

To address the first problem, multivariable regression methods have been proposed which
simultaneously fit many SNPs in a single model [6, 7]. Indeed, the power to detect an associa-
tion for a given SNP may be increased when other causal SNPs have been accounted for. Con-
versely, a stronger signal from a causal SNP may weaken false signals when modeled jointly
[6].

Solutions for confounding by population structure have also received significant attention
in the literature [8–11]. There are two main approaches to account for the relatedness between
subjects: 1) the principal component (PC) adjustment method and 2) the linear mixed model
(LMM). The PC adjustment method includes the top PCs of genome-wide SNP genotypes as

additional covariates in the model [12]. The LMM uses an estimated covariance matrix from the individuals' genotypes and includes this information in the form of a random effect [3].

While these problems have been addressed in isolation, there has been relatively little progress towards addressing them jointly at a large scale. Region-based tests of association have been developed where a linear combination of $p$ variants is regressed on the response variable in a mixed model framework [13]. In case-control data, a stepwise logistic-regression procedure was used to evaluate the relative importance of variants within a small genetic region [14]. These methods however are not applicable in the high-dimensional setting, i.e., when the number of variables $p$ is much larger than the sample size $n$, as is often the case in genetic studies where millions of variants are measured on thousands of individuals.

There has been recent interest in using penalized linear mixed models, which place a constraint on the magnitude of the effect sizes while controlling for confounding factors such as population structure. For example, the LMM-lasso [15] places a Laplace prior on all main effects while the adaptive mixed lasso [16] uses the $L_1$ penalty [17] with adaptively chosen weights [18] to allow for differential shrinkage amongst the variables in the model. Another method applied a combination of both the lasso and group lasso penalties in order to select variants within a gene most associated with the response [19]. However, methods such as the LMM-lasso are normally performed in two steps. First, the variance components are estimated once from a LMM with a single random effect. These LMMs normally use the estimated covariance matrix from the individuals' genotypes to account for the relatedness but assumes no SNP main effects (i.e. a null model). The residuals from this null model with a single random effect can be treated as independent observations because the relatedness has been effectively removed from the original response. In the second step, these residuals are used as the response in any high-dimensional model that assumes uncorrelated errors. This approach has both computational and practical advantages since existing penalized regression software such as `glmnet` [20] and `gglasso` [21], which assume independent observations, can be applied directly to the residuals. However, recent work has shown that there can be a loss in power if a causal variant is included in the calculation of the covariance matrix as its effect will have been removed in the first step [13, 22].

In this paper we develop a general penalized LMM framework called `ggmix` that simultaneously selects variables and estimates their effects, accounting for between-individual correlations. We develop a blockwise coordinate descent algorithm with automatic tuning parameter selection which is highly scalable, computationally efficient and has theoretical guarantees of convergence. Our method can handle several sparsity inducing penalties such as the lasso [17] and elastic net [23]. Through simulations and three real data examples, we show that `ggmix` leads to more parsimonious models compared to the two-stage approach or principal component adjustment with better prediction accuracy. Our method performs well even in the presence of highly correlated markers, and when the causal SNPs are included in the kinship matrix.

All of our algorithms are implemented in the `ggmix` R package hosted on CRAN with extensive documentation (https://sahirbhatnagar.com/ggmix). We provide a brief demonstration of the `ggmix` package in S2 Text.

The rest of the paper is organized as follows. In Results, we compare the performance of our proposed approach and demonstrate the scenarios where it can be advantageous to use over existing methods through simulation studies and three real data analyses. This is followed by a discussion of our results, some limitations and future directions in Discussion. Materials and methods describes the `ggmix` model, the optimization procedure and the algorithm used to fit it.

## Results

In this section we demonstrate the performance of `ggmix` in a simulation study and three real data applications.

### Simulation study

We evaluated the performance of `ggmix` in a variety of simulated scenarios. For each simulation scenario we compared `ggmix` to the `lasso` and the `twostep` method. For the `lasso`, we included the top 10 principal components from the simulated genotypes used to calculate the kinship matrix as unpenalized predictors in the design matrix. For the `twostep` method, we first fitted an intercept only model with a single random effect using the average information restricted maximum likelihood (AIREML) algorithm [24] as implemented in the `gaston` R package [25]. The residuals from this model were then used as the response in a regular `lasso` model. Note that in the `twostep` method, we removed the kinship effect in the first step and therefore did not need to make any further adjustments when fitting the penalized model. We fitted the `lasso` using the default settings and `standardize = FALSE` in the `glmnet` package [20], with 10-fold cross-validation (CV) to select the optimal tuning parameter. For other parameters in our simulation study, we defined the following quantities:

- $n$: sample size

- $c$: percentage of causal SNPs

- $\boldsymbol{\beta}$: true effect size vector of length $p$

- $S_0 = \{j; (\beta)_j \neq 0\}$ the index of the true active set with cardinality $|S_0| = c \times p$

- *causal*: the list of causal SNP indices

- *kinship*: the list of SNP indices for the kinship matrix

- $\mathbf{X}$: $n \times p$ matrix of SNPs that were included as covariates in the model

    We simulated data from the model

$$\mathbf{Y} = \mathbf{X}\boldsymbol{\beta} + \mathbf{P} + \boldsymbol{\varepsilon},$$

where $\mathbf{P} \sim \mathcal{N}(0, \eta\sigma^2\boldsymbol{\Phi})$ is the polygenic effect and $\boldsymbol{\varepsilon} \sim \mathcal{N}(0, (1-\eta)\sigma^2\mathbf{I})$ is the error term. Here, $\boldsymbol{\Phi}_{n \times n}$ is the covariance matrix based on the *kinship* SNPs from $n$ individuals, $\mathbf{I}_{n \times n}$ is the identity matrix and parameters $\sigma^2$ and $\eta \in [0, 1]$ determine how the variance is divided between $\mathbf{P}$ and $\boldsymbol{\varepsilon}$. The values of the parameters that we used were as follows: narrow sense heritability $\eta = \{0.1, 0.3\}$, number of covariates $p = 5{,}000$, number of *kinship* SNPs $k = 10{,}000$, percentage of *causal* SNPs $c = \{0\%, 1\%\}$ and $\sigma^2 = 1$. In addition to these parameters, we also varied the amount of overlap between the *causal* list and the *kinship* list. We considered two main scenarios:

1. None of the *causal* SNPs are included in *kinship* set.

2. All of the *causal* SNPs are included in the *kinship* set.

   Both kinship matrices were meant to contrast the model behavior when the causal SNPs are included in both the main effects and random effects (referred to as proximal contamination [8]) versus when the causal SNPs are only included in the main effects. These scenarios are motivated by the current standard of practice in GWAS where the candidate marker is excluded from the calculation of the kinship matrix [8]. This approach becomes much more difficult to apply in large-scale multivariable models where there is likely to be overlap between

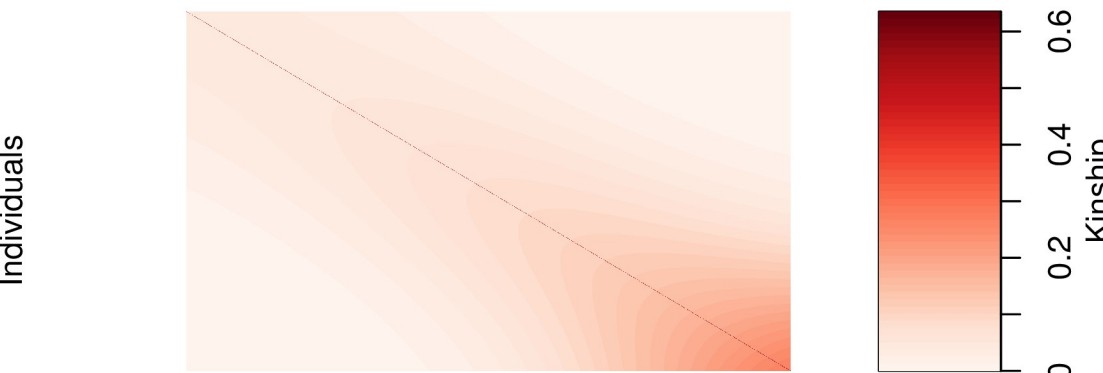

**Fig 1. Empirical kinship matrix.** Example of an empirical kinship matrix used in simulation studies. This scenario models a 1D geography with extensive admixture.

the variables in the design matrix and kinship matrix. We simulated random genotypes from the BN-PSD admixture model with 1D geography and 10 subpopulations using the `bnpsd` package [26, 27]. In Fig 1, we plot the estimated kinship matrix from a single simulated dataset in the form of a heatmap where a darker color indicates a closer genetic relationship.

In Fig 2 we plot the first two principal component scores calculated from the simulated genotypes used to calculate the kinship matrix in Fig 1, and color each point by subpopulation membership. We can see that the PCs can identify the subpopulations which is why including them as additional covariates in a regression model has been considered a reasonable approach to control for confounding.

Using this set-up, we randomly partitioned 1000 simulated observations into 80% for training and 20% for testing. The training set was used to fit the model and select the optimal tuning parameter only, and the resulting model was evaluated on the test set. Let $\hat{\lambda}$ be the estimated value of the optimal regularization parameter, $\hat{\boldsymbol{\beta}}_{\hat{\lambda}}$ the estimate of $\boldsymbol{\beta}$ at regularization parameter $\hat{\lambda}$, and $\hat{S}_{\hat{\lambda}} = \{j; (\hat{\boldsymbol{\beta}}_{\hat{\lambda}})_j \neq 0\}$ the index of the set of non-zero estimated coefficients.

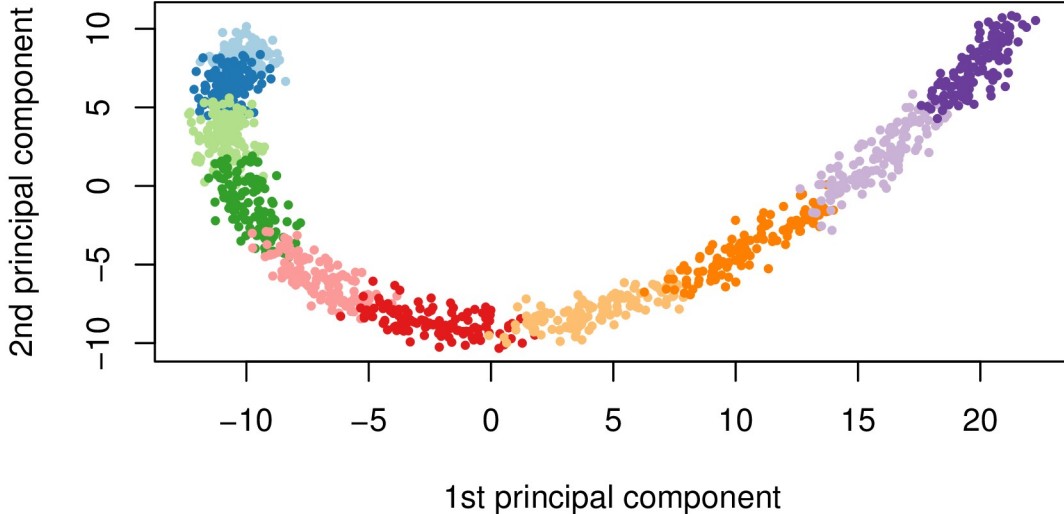

**Fig 2. First two principal components.** First two principal component scores of the genotype data used to estimate the kinship matrix where each color represents one of the 10 simulated subpopulations.

**Table 1. Simulation study results.** Mean (standard deviation) from 200 simulations stratified by the number of causal SNPs (null, 1%), the overlap between causal SNPs and kinship matrix (no overlap, all causal SNPs in kinship), and true heritability (10%, 30%). For all simulations, sample size is $n = 1000$, the number of covariates is $p = 5000$, and the number of SNPs used to estimate the kinship matrix is $k = 10000$. TPR at FPR = 5% is the true positive rate at a fixed false positive rate of 5%. Model Size ($|\hat{S}_{\hat{\lambda}}|$) is the number of selected variables in the training set using the high-dimensional BIC for `ggmix` and 10-fold cross validation for `lasso` and `twostep`. RMSE is the root mean squared error on the test set. Estimation error is the squared distance between the estimated and true effect sizes. Error variance ($\sigma^2$) for `twostep` is estimated from an intercept only LMM with a single random effect and is modeled explicitly in `ggmix`. For the `lasso` we use $\frac{1}{n-|\hat{S}_{\hat{\lambda}}|}\|\mathbf{Y} - \mathbf{X}\hat{\boldsymbol{\beta}}_{\hat{\lambda}}\|_2^2$ [28] as an estimator for $\sigma^2$. Heritability ($\eta$) for `twostep` is estimated as $\sigma_g^2/(\sigma_g^2 + \sigma_e^2)$ from an intercept only LMM with a single random effect where $\sigma_g^2$ and $\sigma_e^2$ are the variance components for the random effect and error term, respectively. $\eta$ is explictly modeled in `ggmix`. There is no positive way to calculate $\eta$ for the `lasso` since we are using a PC adjustment.

| | | Null model | | | | 1% Causal SNPs | | | |
| | | No overlap | | All causal SNPs in kinship | | No overlap | | All causal SNPs in kinship | |
| Metric | Method | 10% | 30% | 10% | 30% | 10% | 30% | 10% | 30% |
|---|---|---|---|---|---|---|---|---|---|
| **TPR at FPR** | twostep | 0.00 (0.00) | 0.00 (0.00) | 0.00 (0.00) | 0.00 (0.00) | 0.84 (0.05) | 0.84 (0.05) | 0.76 (0.09) | 0.77 (0.08) |
| | lasso | 0.00 (0.00) | 0.00 (0.00) | 0.00 (0.00) | 0.00 (0.00) | 0.86 (0.05) | 0.85 (0.05) | 0.86 (0.05) | 0.86 (0.05) |
| | ggmix | 0.00 (0.00) | 0.00 (0.00) | 0.00 (0.00) | 0.00 (0.00) | 0.86 (0.05) | 0.86 (0.05) | 0.85 (0.05) | 0.86 (0.05) |
| **Model Size** | twostep | 0 (0, 5) | 0 (0, 2) | 0 (0, 5) | 0 (0, 2) | 328 (289, 388) | 332 (287, 385) | 284 (250, 329) | 284 (253, 319) |
| | lasso | 0 (0, 6) | 0 (0, 5) | 0 (0, 6) | 0 (0, 5) | 278 (246, 317) | 276 (245, 314) | 279 (252, 321) | 285 (244, 319) |
| | ggmix | 0 (0, 0) | 0 (0, 0) | 0 (0, 0) | 0 (0, 0) | 43 (39, 49) | 43 (39, 48) | 44 (38, 49) | 43 (38, 48) |
| **RMSE** | twostep | 1.02 (0.07) | 1.02 (0.06) | 1.02 (0.07) | 1.02 (0.06) | 1.42 (0.10) | 1.41 (0.10) | 1.44 (0.33) | 1.40 (0.22) |
| | lasso | 1.02 (0.06) | 1.02 (0.06) | 1.02 (0.06) | 1.02 (0.06) | 1.39 (0.09) | 1.38 (0.09) | 1.40 (0.08) | 1.38 (0.08) |
| | ggmix | 1.00 (0.05) | 1.00 (0.05) | 1.00 (0.05) | 1.00 (0.05) | 1.22 (0.10) | 1.20 (0.10) | 1.23 (0.11) | 1.23 (0.12) |
| **Estimation Error** | twostep | 0.12 (0.22) | 0.09 (0.19) | 0.12 (0.22) | 0.09 (0.19) | 2.97 (0.60) | 2.92 (0.60) | 3.60 (5.41) | 3.21 (3.46) |
| | lasso | 0.13 (0.21) | 0.12 (0.22) | 0.13 (0.21) | 0.12 (0.22) | 2.76 (0.46) | 2.69 (0.47) | 2.82 (0.48) | 2.75 (0.48) |
| | ggmix | 0.00 (0.01) | 0.01 (0.02) | 0.00 (0.01) | 0.01 (0.02) | 2.11 (1.28) | 2.04 (1.22) | 2.21 (1.24) | 2.28 (1.34) |
| **Error Variance** | twostep | 0.87 (0.11) | 0.69 (0.15) | 0.87 (0.11) | 0.69 (0.15) | 14.23 (3.53) | 14.13 (3.52) | 1.42 (1.71) | 1.28 (1.66) |
| | lasso | 0.98 (0.05) | 0.96 (0.05) | 0.98 (0.05) | 0.96 (0.05) | 1.04 (0.13) | 1.02 (0.13) | 1.03 (0.14) | 1.01 (0.14) |
| | ggmix | 0.85 (0.18) | 0.64 (0.20) | 0.85 (0.18) | 0.64 (0.20) | 2.00 (0.49) | 1.86 (0.51) | 1.06 (0.46) | 0.83 (0.45) |
| **Heritability** | twostep | 0.13 (0.11) | 0.31 (0.15) | 0.13 (0.11) | 0.31 (0.15) | 0.26 (0.14) | 0.26 (0.14) | 0.92 (0.08) | 0.93 (0.08) |
| | lasso | – | – | – | – | – | – | – | – |
| | ggmix | 0.15 (0.18) | 0.37 (0.21) | 0.15 (0.18) | 0.37 (0.21) | 0.18 (0.16) | 0.23 (0.17) | 0.59 (0.20) | 0.68 (0.19) |

Note: median (inter-quartile range) is given for model size.

To compare the methods in the context of true positive rate (TPR), we selected the largest tuning parameter that would result in a false positive rate (FPR) closest to 5%, but not more. Note that in practice, this approach to selecting the tuning parameter is generally not possible since we do not know the underlying true model in advance. For real data, we suggest an information criterion approach described in S1 Text or a sample splitting approach such as the one we used for the UK Biobank analysis. We also compared the model size ($|\hat{S}_{\hat{\lambda}}|$), test set prediction error based on the refitted unpenalized estimates for each selected model, the estimation error ($\|\hat{\boldsymbol{\beta}} - \boldsymbol{\beta}\|_2^2$), and the variance components ($\eta, \sigma^2$) for the polygenic random effect and error term.

The results are summarized in Table 1. We see that `ggmix` outperformed the `twostep` in terms of TPR, and was comparable to the `lasso`. This was the case, regardless of true heritability and whether the causal SNPs were included in the calculation of the kinship matrix. For the `twostep` however, the TPR at a FPR of 5%, drops, on average, from 0.84 (when causal SNPs are not in the kinship) to 0.76 (when causal SNPs are in the kinship). Across all simulation scenarios, `ggmix` had the smallest estimation error, and smallest root mean squared prediction error (RMSE) on the test set while also producing the most parsimonious models. Both the `lasso` and `twostep` selected more false positives, even in the null model scenario. Both

the `twostep` and `ggmix` overestimated the heritability though `ggmix` was closer to the true value. When none of the causal SNPs were in the kinship, both methods tended to overestimate the truth when $\eta$ = 10% and underestimate when $\eta$ = 30%. Across all simulation scenarios `ggmix` was able to (on average) correctly estimate the error variance. The `lasso` tended to overestimate $\sigma^2$ in the null model while the `twostep` overestimated $\sigma^2$ when none of the causal SNPs were in the kinship matrix.

Overall, we observed that variable selection results and RMSE for `ggmix` were similar regardless of whether the causal SNPs were in the kinship matrix or not. This result is encouraging since in practice the kinship matrix is constructed from a random sample of SNPs across the genome, some of which are likely to be causal, particularly in polygenic traits.

In particular, our simulation results show that the principal component adjustment method may not be the best approach to control for confounding by population structure, particularly when variable selection is of interest.

## Real data applications

Three datasets with different features were used to illustrate the potential advantages of `ggmix` over existing approaches such as PC adjustment in a `lasso` regression. In the first two datasets, family structure induced low levels of correlation and sparsity in signals. In the last, a dataset involving mouse crosses, correlations were extremely strong and could confound signals.

**UK Biobank.**   With more than 500,000 participants, the UK Biobank is one of the largest genotyped health care registries in the world. Among these participants, 147,731 have been inferred to be related to at least one individual in this cohort [29]. Such a widespread genetic relatedness may confound association studies and bias trait predictions if not properly accounted for. Among these related individuals, 18,150 have a documented familial relationship (parent-offspring, full siblings, second degree or third degree) that was previously inferred in [30]. We attempted to derive a polygenic risk score for height among these individuals. As suggested by a reviewer, the goal of this analysis was to see how the different methods performed for a highly polygenic trait in a set of related individuals. We compared the `ggmix`-derived polygenic risk score to those derived by the `twostep` and `lasso` methods.

We first estimated the pairwise kinship coefficient among the 18,150 reportedly related individuals based on 784,256 genotyped SNPs using KING [31]. We grouped related individuals with a kinship coefficient > 0.044 [31] into 8,300 pedigrees. We then randomly split the dataset into a training set, a model selection set and a test set of roughly equal sample size, ensuring all individuals in the same pedigree were assigned into the same set. We inverse normalized the standing height after adjusting for age, sex, genotyping array, and assessment center following Yengo et al. [32].

To reduce computational complexity, we selected 10,000 SNPs with the largest effect sizes associated with height from a recent large meta-analysis [32]. Among these 10,000 SNPs, 1,233 were genotyped and used for estimating the kinship whereas the other 8,767 SNPs were imputed based on the Haplotype Reference Consortium reference panel [33]. The distribution of the 10,000 SNPs by chromosome and whether or not the SNP was imputed is shown in S1 Fig. We see that every chromosome contributed SNPs to the model with 15% coming from chromosome 6. The markers we used are theoretically independent since Yengo et al. performed a COJO analysis which should have tuned down signals due to linkage disequilibrium [32]. We used `ggmix`, `twostep` and `lasso` to select SNPs most predictive of the inverse normalized height on the training set, and chose the λ with the lowest prediction RMSE on the model selection set for each method. We then examined the performance of each derived

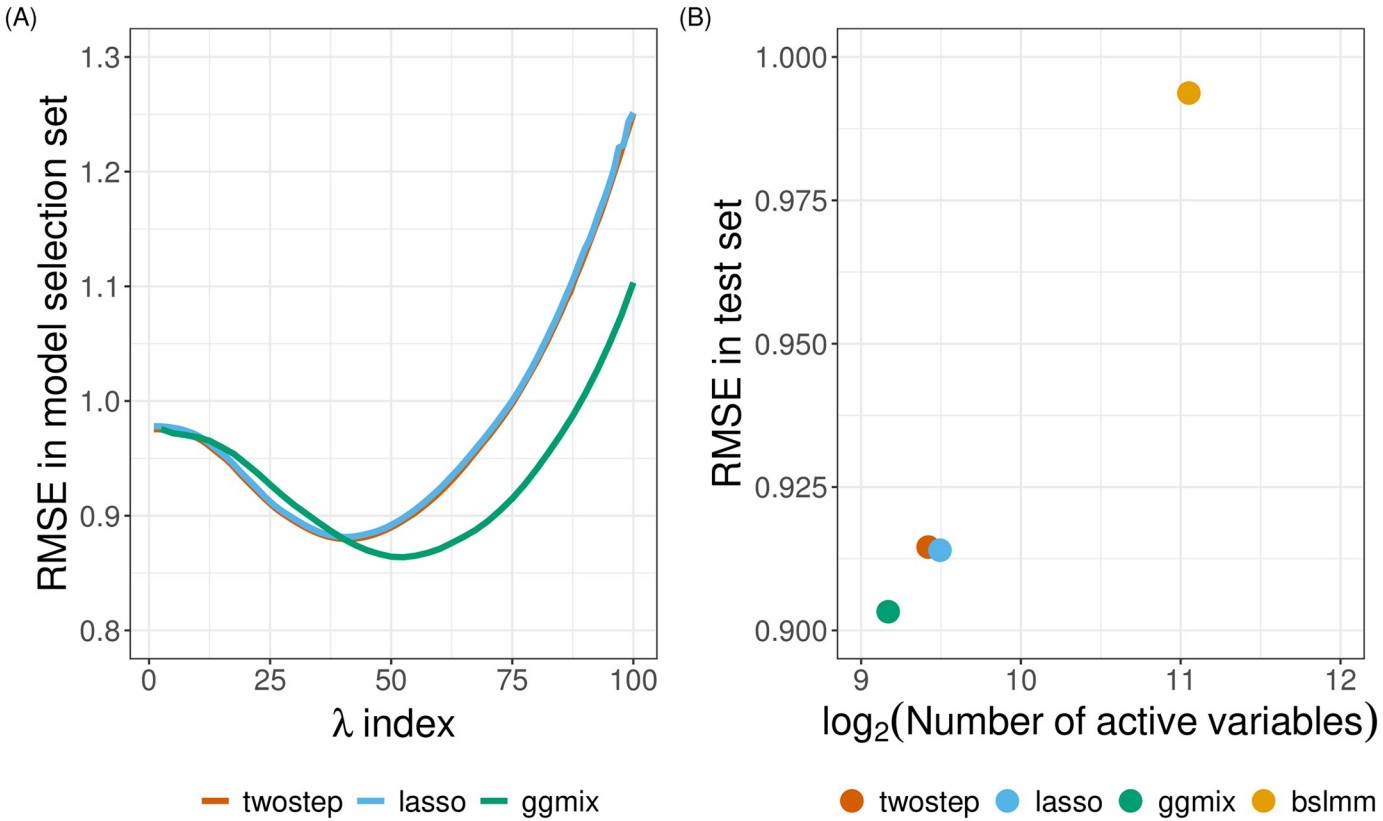

**Fig 3. Model selection and testing in the UK Biobank.** (**a**) Root-mean-square error of three methods on the model selection set with respect to a grid search of penalty factor used on the training set. (**b**) Performance of four methods on the test set with penalty factor optimized on the model selection set. The x-axis has a logarithmic scale. The BSLMM method optimized coefficients of each SNP through an MCMC process on the training set and was directly evaluated on the test set.

polygenic risk score on the test set. Similar to simulation study, we adjusted for the top 10 genetic PCs as unpenalized predictors when fitting the `lasso` models, and supplied the kinship matrix based on 784,256 genotyped SNPs to `ggmix` and `twostep`.

We found that with a kinship matrix estimated using all genotyped SNPs, `ggmix` had the possibility to achieve a lower RMSE on the model selection set compared to the `twostep` and `lasso` methods (Fig 3A). An optimized `ggmix`-derived polygenic risk score that utilized the least number of SNPs was also able to better predict the trait with lower RMSE on the test set (Fig 3B).

We additionally applied a Bayesian Sparse Linear Mixed Model (`BSLMM`) [34] implemented in the GEMMA package [35] to derive a polygenic risk score on the training set. A posterior probability of inclusion of each SNP was provided and prediction was based on all SNPs with a positive posterior probability. We found that although the `BSLMM`-based polygenic risk score leveraged the most SNPs, it did not achieve a comparable prediction accuracy as the other three methods (Fig 3B). Likely due to the small effect sizes of these SNPs, only 94, 35 and 1 SNPs had a posterior inclusion probability above 0.05, 0.10 and 0.50, respectively. The model would have further reduced prediction accuracy if the prediction was based only on these SNPs.

**GAW20.** In the most recent Genetic Analysis Workshop 20 (GAW20), the causal modeling group investigated causal relationships between DNA methylation (exposure) within some genes and the change in high-density lipoproteins ΔHDL (outcome) using Mendelian

Randomization (MR) [36]. Penalized regression methods were used to select SNPs strongly associated with the exposure in order to be used as an instrumental variable (IV) [37, 38]. However, since GAW20 data consisted of families, `twostep` methods were used which could have resulted in a large number of false positives or false negatives. `ggmix` now provides an alternative approach that could be used for selecting the IV while accounting for the family structure of the data.

We applied `ggmix` to all 200 GAW20 simulation datasets, each of 679 observations, and compared its performance to the `twostep` and `lasso` methods. Using a Factored Spectrally Transformed Linear Mixed Model (FaST-LMM) [39] adjusted for age and sex, we validated the effect of rs9661059 on blood lipid trait to be significant (genome-wide $p = 6.29 \times 10^{-9}$). Though several other SNPs were also associated with the phenotype, these associations were probably mediated by CpG-SNP interaction pairs and did not reach statistical significance. Therefore, to avoid ambiguity, we only focused on chromosome 1 containing 51,104 SNPs, including rs9661059. Given that population admixture in the GAW20 data was likely, we estimated the population kinship using REAP [40] after decomposing population compositions using ADMIXTURE [41]. We used 100,276 LD-pruned whole-genome genotyped SNPs for estimating the kinship. Among these, 8100 were included as covariates in our models based on chromosome 1. The causal SNP was also among the 100,276 SNPs. All methods were fit according to the same settings described in our simulation study, and adjusting for age and sex. We calculated the median (inter-quartile range) number of active variables, and RMSE (standard deviation) based on five-fold CV on each simulated dataset.

On each simulated replicate, we calibrated the methods so that they could be easily compared by fixing the true positive rate to 1 and then minimizing the false positive rate. Hence, the selected SNP, rs9661059, was likely to be the true positive for each method, and non-causal SNPs were excluded to the greatest extent. All three methods precisely chose the correct predictor without any false positives in more than half of the replicates, as the causal signal was strong. However, when some false positives were selected (i.e. when the number of active variables > 1), `ggmix` performed comparably to `twostep`, while the `lasso` was inclined to select more false positives as suggested by the larger third quartile number of active variables (Table 2). We also observed that `ggmix` outperformed the `twostep` method with lower CV RMSE using the same number of SNPs. Meanwhile, it achieved roughly the same prediction accuracy as `lasso` but with fewer non-causal SNPs (Table 2). It is also worth mentioning that

**Table 2. GAW20 simulation study results.** Summary of model performance based on 200 GAW20 simulations for the `twostep`, `lasso`, `ggmix` and `BSLMM` model with different posterior inclusion probability (PIP) thresholds. Five-fold cross-validation root-mean-square error (RMSE) was reported for each simulation replicate. Prediction performance was not reported for `BSLMM` with PIP greater than 0.05, 0.10 and 0.50 because some of the replications contained no active SNPs.

| Method | Model Size | RMSE (SD) |
|---|---|---|
| `twostep` | 1 (1–11) | 0.3604 (0.0242) |
| `lasso` | 1 (1–15) | 0.3105 (0.0199) |
| `ggmix` | 1 (1–12) | 0.3146 (0.0210) |
| `BSLMM (PIP > 0)` | 40,737 (39,901–41,539) | 0.2503 (0.0099) |
| `BSLMM (PIP > 0.05)` | 2 (1–4) | - |
| `BSLMM (PIP > 0.10)` | 0 (0–1) | - |
| `BSLMM (PIP > 0.50)` | 0 (0–0) | - |

Note: median (inter-quartile range) is given for model size.

there was very little correlation between the causal SNP and SNPs within a 1Mb-window around it (see S2 Fig), making it an ideal scenario for the `lasso` and related methods.

We also applied the `BSLMM` method by performing five-fold CV on each of the 200 simulated replicates. We found that while `BSLMM` achieved a lower CV RMSE compared to the other methods (Table 2), this higher prediction accuracy relied on approximately 80% of the 51,104 SNPs with a positive posterior inclusion probability. This may suggest overfitting in this dataset. We additionally tried imposing a stricter posterior inclusion probability threshold (0.05, 0.10 and 0.50) in order to improve feature selection. These thresholds however, resulted in overly sparse models as most SNPs had a low posterior probability. It is also noteworthy that we did not adjust for age and sex in the `BSLMM` model, as the current implementation of the method in the GEMMA package does not allow adjustment for covariates.

**Mouse crosses and sensitivity to mycobacterial infection.**   Mouse inbred strains of genetically identical individuals are extensively used in research. Crosses of different inbred strains are useful for various studies of heritability focusing on either observable phenotypes or molecular mechanisms, and in particular, recombinant congenic strains have been an extremely useful resource for many years [42]. However, ignoring complex genetic relationships in association studies can lead to inflated false positives in genetic association studies when different inbred strains and their crosses are investigated [43–45]. Therefore, a previous study developed and implemented a mixed model to find loci associated with mouse sensitivity to mycobacterial infection [46]. The random effects in the model captured complex correlations between the recombinant congenic mouse strains based on the proportion of the DNA shared identical by descent. Through a series of mixed model fits at each marker, new loci that impact growth of mycobacteria on chromosome 1 and chromosome 11 were identified.

Here we show that `ggmix` can identify these loci, as well as potentially others, in a single analysis. We reanalyzed the growth permissiveness in the spleen, as measured by colony forming units (CFUs), 6 weeks after infection from *Mycobacterium bovis* Bacille Calmette-Guerin (BCG) Russia strain as reported in [46].

By taking the consensus between the "main model" and the "conditional model" of the original study, we regarded markers D1Mit435 on chromosome 1 and D11Mit119 on chromosome 11 as two true positive loci. We directly estimated the kinship between mice using genotypes at 625 microsatellite markers. The estimated kinship was entered directly into `ggmix` and `twostep`. For the `lasso`, we calculated and included the first 10 principal components of the estimated kinship. To evaluate the robustness of different models, we bootstrapped the 189-sample dataset and repeated the analysis 200 times. We then conceived a two-fold criteria to evaluate performance of each model. We first examined whether a model could pick up both true positive loci using some λ. If the model failed to pick up both loci simultaneously with any λ, we counted as modeling failure on the corresponding boostrap replicate; otherwise, we counted as modeling success and recorded which other loci were picked up given the largest λ. Consequently, similar to the strategy used in the GAW20 analysis, we optimized the models by tuning the penalty factor such that these two true positive loci were picked up, while the number of other active loci was minimized. Significant markers were defined as those captured in at least half of the successful bootstrap replicates (Fig 4).

We demonstrated that `ggmix` recognized the true associations more robustly than `twostep` and `lasso`. In almost all (99%) bootstrap replicates, `ggmix` was able to capture both true positives, while the `twostep` failed in 19% of the replicates and the `lasso` failed in 56% of the replicates by missing at least one of the two true positives (Fig 4). The robustness of `ggmix` is particularly noteworthy due to the strong correlations between all microsatellite markers in this dataset (see S3 Fig). These strong correlations with the causal markers, partially

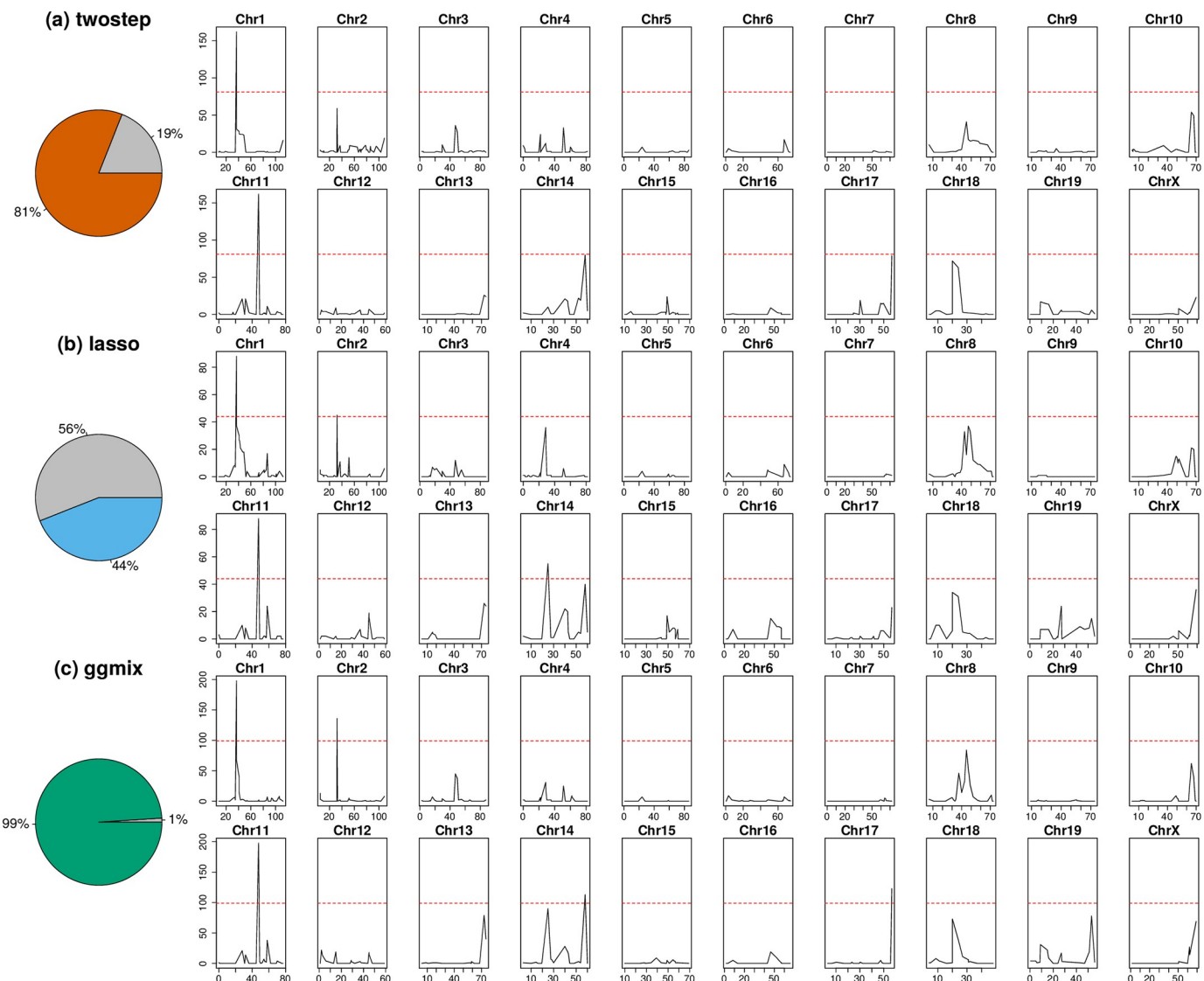

**Fig 4. Comparison of model performance on the mouse cross data.** Pie charts depict model robustness where grey areas denote bootstrap replicates on which the corresponding model is unable to capture both true positives using any penalty factor, whereas colored areas denote successful replicates. Chromosome-based signals record in how many successful replicates the corresponding loci are picked up by the corresponding optimized model. Red dashed lines delineate significance thresholds.

explain the poor performance of the `lasso` as it suffers from unstable selections in the presence of correlated variables (e.g. [47]).

We also identified several other loci that might also be associated with susceptibility to mycobacterial infection (Table 3). Among these new potentially-associated markers, D2Mit156 was found to play a role in control of parasite numbers of *Leishmania tropica* in lymph nodes [48]. An earlier study identified a parent-of-origin effect at D17Mit221 on CD4M levels [49]. This effect was more visible in crosses than in parental strains. In addition, D14Mit131, selected only by `ggmix`, was found to have a 9% loss of heterozygosity in hybrids of two inbred mouse strains [50], indicating the potential presence of putative suppressor genes pertaining to immune surveillance and tumor progression [51]. This result might also suggest association with anti-bacterial responses yet to be discovered.

**Table 3. Mouse crosses and sensitivity to mycobacterial infection.** Additional loci significantly associated with mouse susceptibility to mycobacterial infection, after excluding two true positives. Loci needed to be identified in at least 50% of the successful bootstrap replicates that captured both true positive loci.

| Method | Marker | Position in cM | Position in bp |
|---|---|---|---|
| twostep | N/A | N/A | N/A |
| lasso | D2Mit156 | Chr2:31.66 | Chr2:57081653-57081799 |
| | D14Mit155 | Chr14:31.52 | Chr14:59828398-59828596 |
| ggmix | D2Mit156 | Chr2:31.66 | Chr2:57081653-57081799 |
| | D14Mit131 | Chr14:63.59 | Chr14:120006565-120006669 |
| | D17Mit221 | Chr17:59.77 | Chr17:90087704-90087842 |

Note: median (inter-quartile range) is given for model size.

## Discussion

We have developed a general penalized LMM framework called ggmix which simultaneously selects SNPs and adjusts for population structure in high dimensional prediction models. We compared our method to the twostage procedure, where in the first stage, the dependence between observations is adjusted for in a LMM with a single random effect and no covariates (i.e. null model). The residuals from this null model can then be used in any model for independent observations because the relatedness has been effectively removed from the original response. We also compared our method to the lasso and BSLMM which are closely related to ggmix since they also jointly model the relatedness and SNPs in a single step. The key differences are that the lasso uses a principal component adjustment and BSLMM is a Bayesian method focused on phenotype prediction.

Through an extensive simulation study and three real data analyses that mimic many experimental designs in genetics, we show that the current approaches of PC adjustment and two-stage procedures are not necessarily sufficient to control for confounding by population structure leading to a high number of false positives. Our simulation results show that ggmix outperforms existing methods in terms of sparsity and prediction error even when the causal variants are included in the kinship matrix (Table 1). Many methods for single-SNP analyses avoid this proximal contamination [8] by using a leave-one-chromosome-out scheme [52], i.e., construct the kinship matrix using all chromosomes except the one on which the marker being tested is located. However, this approach is not possible if we want to model many SNPs (across many chromosomes) jointly to create, for example, a polygenic risk score. For the purposes of variable selection, we would also want to model all chromosomes together since the power to detect an association for a given SNP may be increased when other causal SNPs have been accounted for. Conversely, a stronger signal from a causal SNP may weaken false signals when modeled jointly [6], particularly when the markers are highly correlated as in the mouse crosses example.

In the UK Biobank, we found that with a kinship matrix estimated using all genotyped SNPs, ggmix had achieved a lower RMSE on the model selection set compared to the twostep and lasso methods. Furthermore, an optimized ggmix-derived polygenic risk score that utilized the least number of SNPs was also able to better predict the trait with lower RMSE on the test set. In the GAW20 example, we showed that while all methods were able to select the strongest causal SNP, ggmix did so with the least amount of false positives while also maintaining good predictive ability. In the mouse crosses example, we showed that ggmix is robust to perturbations in the data using a bootstrap analysis. Indeed, ggmix was able to consistently select the true positives across bootstrap replicates, while twostep failed in 19% of the replicates and lasso failed in 56% of the replicates by missing of at least one of the two

true positives. Our re-analysis of the data also lead to some potentially new findings, not found by existing methods, that may warrant further study. This particular example had many markers that were strongly correlated with each other (see S3 Fig). Nevertheless, we observed that the two true positive loci were the most often selected while none of the nearby markers were picked up in more than 50% of the 200 bootstrap replicates. This shows that our method does recognize the true positives in the presence of highly correlated markers. Nevertheless, we think the issue of variable selection for correlated SNPs warrants further study. The recently proposed Precision Lasso [47] seeks to address this problem in the high-dimensional fixed effects model.

We emphasize here that previously developed methods such as the LMM-lasso [15] use a two-stage fitting procedure without any convergence details. From a practical point of view, there is currently no implementation that provides a principled way of determining the sequence of tuning parameters to fit, nor a procedure that automatically selects the optimal value of the tuning parameter. To our knowledge, we are the first to develop a coordinate gradient descent (CGD) algorithm in the specific context of fitting a penalized LMM for population structure correction with theoretical guarantees of convergence. Furthermore, we develop a principled method for automatic tuning parameter selection and provide an easy-to-use software implementation in order to promote wider uptake of these more complex methods by applied practitioners.

Although we derive a CGD algorithm for the $\ell_1$ penalty, our approach can also be easily extended to other penalties such as the elastic net and group lasso with the same guarantees of convergence. A limitation of `ggmix` is that it first requires computing the covariance matrix with a computation time of $\mathcal{O}(n^2 k)$ followed by a spectral decomposition of this matrix in $\mathcal{O}(n^3)$ time where $k$ is the number of SNP genotypes used to construct the covariance matrix. This computation becomes prohibitive for large cohorts such as the UK Biobank [53] which have collected genetic information on half a million individuals. When the matrix of genotypes used to construct the covariance matrix is low rank, there are additional computational speedups that can be implemented. While this has been developed for the univariate case [8], to our knowledge, this has not been explored in the multivariable case. We are currently developing a low rank version of the penalized LMM developed here, which reduces the time complexity from $\mathcal{O}(n^2 k)$ to $\mathcal{O}(nk^2)$. There is also the issue of how our model scales with an increasing number of covariates ($p$). Due to the coordinate-wise optimization procedure, we expect this to be less of an issue, but still prohibitive for $p > 1 \times 10^5$. The `biglasso` package [54] uses memory mapping strategies for large $p$, and this is something we are exploring for `ggmix`.

As was brought up by a reviewer, the simulations and real data analyses presented here contained many more markers used to estimate the kinship than the sample size ($n/k \leq 0.1$). In the single locus association test, Yang el al. [22] found that proximal contamination was an issue when $n/k \approx 1$. We believe further theoretical study is needed to see if these results can be generalized to the multivariable models being fit here. Once the computational limitations of sample size mentioned above have been addressed, these theoretical results can be supported by simulation studies.

There are other applications in which our method could be used as well. For example, there has been a renewed interest in polygenic risk scores (PRS) which aim to predict complex diseases from genotypes. `ggmix` could be used to build a PRS with the distinct advantage of modeling SNPs jointly, allowing for main effects as well as interactions to be accounted for. Based on our results, `ggmix` has the potential to produce more robust and parsimonious models than the `lasso` with better predictive accuracy.

Our method is also suitable for fine mapping SNP association signals in genomic regions, where the goal is to pinpoint individual variants most likely to impact the underlying biological mechanisms of disease [55].

## Materials and methods

### Model set-up

Let $i = 1, \ldots, N$ be a grouping index, $j = 1, \ldots, n_i$ the observation index within a group and $N_T = \sum_{i=1}^{N} n_i$ the total number of observations. For each group let $\boldsymbol{y}_i = (y_1, \ldots, y_{n_i})$ be the observed vector of responses or phenotypes, $\mathbf{X}_i$ an $n_i \times (p + 1)$ design matrix (with the column of 1s for the intercept), $\boldsymbol{b}_i$ a group-specific random effect vector of length $n_i$ and $\boldsymbol{\varepsilon}_i = (\varepsilon_{i1}, \ldots, \varepsilon_{in_i})$ the individual error terms. Denote the stacked vectors $\mathbf{Y} = (\boldsymbol{y}_i, \ldots, \boldsymbol{y}_N)^T \in \mathbb{R}^{N_T \times 1}, \boldsymbol{b} = (\boldsymbol{b}_i, \ldots, \boldsymbol{b}_N)^T \in \mathbb{R}^{N_T \times 1}, \boldsymbol{\varepsilon} = (\boldsymbol{\varepsilon}_i, \ldots, \boldsymbol{\varepsilon}_N)^T \in \mathbb{R}^{N_T \times 1}$, and the stacked matrix $\mathbf{X} = (\mathbf{X}_1^T, \ldots, \mathbf{X}_N^T) \in \mathbb{R}^{N_T \times (p+1)}$. Furthermore, let $\boldsymbol{\beta} = (\boldsymbol{\beta}_0, \boldsymbol{\beta}_1, \ldots, \boldsymbol{\beta}_p)^T \in \mathbb{R}^{(p+1) \times 1}$ be a vector of fixed effects regression coefficients corresponding to $\mathbf{X}$. We consider the following linear mixed model with a single random effect [56]:

$$\mathbf{Y} = \mathbf{X}\boldsymbol{\beta} + \boldsymbol{b} + \boldsymbol{\varepsilon},$$

where the random effect $\boldsymbol{b}$ and the error variance $\boldsymbol{\varepsilon}$ are assigned the distributions

$$\boldsymbol{b} \sim \mathcal{N}(0, \eta\sigma^2\boldsymbol{\Phi}) \qquad \boldsymbol{\varepsilon} \sim \mathcal{N}(0, (1 - \eta)\sigma^2\mathbf{I}).$$

Here, $\boldsymbol{\Phi}_{N_T \times N_T}$ is a known positive semi-definite and symmetric covariance or kinship matrix calculated from SNPs sampled across the genome, $\mathbf{I}_{N_T \times N_T}$ is the identity matrix and parameters $\sigma^2$ and $\eta \in [0, 1]$ determine how the variance is divided between $\boldsymbol{b}$ and $\boldsymbol{\varepsilon}$. Note that $\eta$ is also the narrow-sense heritability ($h^2$), defined as the proportion of phenotypic variance attributable to the additive genetic factors [1]. The joint density of $\mathbf{Y}$ is therefore multivariate normal:

$$\mathbf{Y}|(\boldsymbol{\beta}, \eta, \sigma^2) \sim \mathcal{N}(\mathbf{X}\boldsymbol{\beta}, \eta\sigma^2\boldsymbol{\Phi} + (1 - \eta)\sigma^2\mathbf{I}). \tag{1}$$

The LMM-Lasso method [15] considers an alternative but equivalent parameterization given by:

$$\mathbf{Y}|(\boldsymbol{\beta}, \delta, \sigma_g^2) \sim \mathcal{N}(\mathbf{X}\boldsymbol{\beta}, \sigma_g^2(\boldsymbol{\Phi} + \delta\mathbf{I})), \tag{2}$$

where $\delta = \sigma_e^2/\sigma_g^2$, $\sigma_g^2$ is the genetic variance and $\sigma_e^2$ is the residual variance. We instead consider the parameterization in Eq 1 since maximization is easier over the compact set $\eta \in [0, 1]$ than over the unbounded interval $\delta \in [0, \infty)$ [56]. We define the complete parameter vector as $\boldsymbol{\Theta} := (\boldsymbol{\beta}, \eta, \sigma^2)$. The negative log-likelihood for Eq 1 is given by

$$-\ell(\boldsymbol{\Theta}) \propto \frac{N_T}{2}\log(\sigma^2) + \frac{1}{2}\log(\det(\mathbf{V})) + \frac{1}{2\sigma^2}(\mathbf{Y} - \mathbf{X}\boldsymbol{\beta})^T\mathbf{V}^{-1}(\mathbf{Y} - \mathbf{X}\boldsymbol{\beta}), \tag{3}$$

where $\mathbf{V} = \eta\boldsymbol{\Phi} + (1 - \eta)\mathbf{I}$ and $\det(\mathbf{V})$ is the determinant of $\mathbf{V}$.

Let $\boldsymbol{\Phi} = \mathbf{U}\mathbf{D}\mathbf{U}^T$ be the eigen (spectral) decomposition of the kinship matrix $\boldsymbol{\Phi}$, where $\mathbf{U}_{N_T \times N_T}$ is an orthonormal matrix of eigenvectors (i.e. $\mathbf{U}\mathbf{U}^T = \mathbf{I}$) and $\mathbf{D}_{N_T \times N_T}$ is a diagonal matrix

of eigenvalues $\Lambda_i$. $\mathbf{V}$ can then be further simplified [56]

$$
\begin{aligned}
\mathbf{V} &= \eta\mathbf{\Phi} + (1-\eta)\mathbf{I} \\
&= \eta\mathbf{U}\mathbf{D}\mathbf{U}^T + (1-\eta)\mathbf{U}\mathbf{I}\mathbf{U}^T \\
&= \mathbf{U}\eta\mathbf{D}\mathbf{U}^T + \mathbf{U}(1-\eta)\mathbf{I}\mathbf{U}^T \\
&= \mathbf{U}(\eta\mathbf{D} + (1-\eta)\mathbf{I})\mathbf{U}^T \\
&= \mathbf{U}\tilde{\mathbf{D}}\mathbf{U}^T,
\end{aligned}
\tag{4}
$$

where

$$
\begin{aligned}
\tilde{\mathbf{D}} &= \eta\mathbf{D} + (1-\eta)\mathbf{I} \\
&= \eta
\begin{bmatrix}
\Lambda_1 & & & \\
& \Lambda_2 & & \\
& & \ddots & \\
& & & \Lambda_{N_T}
\end{bmatrix}
+ (1-\eta)
\begin{bmatrix}
1 & & & \\
& 1 & & \\
& & \ddots & \\
& & & 1
\end{bmatrix} \\
&=
\begin{bmatrix}
1 + \eta(\Lambda_1 - 1) & & & \\
& 1 + \eta(\Lambda_2 - 1) & & \\
& & \ddots & \\
& & & 1 + \eta(\Lambda_{N_T} - 1)
\end{bmatrix}
\end{aligned}
\tag{5}
$$

$$
= \mathrm{diag}\{1 + \eta(\Lambda_1 - 1), 1 + \eta(\Lambda_2 - 1), \ldots, 1 + \eta(\Lambda_{N_T} - 1)\}.
\tag{6}
$$

Since Eq 5 is a diagonal matrix, its inverse is also a diagonal matrix:

$$
\tilde{\mathbf{D}}^{-1} = \mathrm{diag}\left\{\frac{1}{1 + \eta(\Lambda_1 - 1)}, \frac{1}{1 + \eta(\Lambda_2 - 1)}, \ldots, \frac{1}{1 + \eta(\Lambda_{N_T} - 1)}\right\}.
\tag{7}
$$

From Eqs 4 and 6, $\log(\det(\mathbf{V}))$ simplifies to

$$
\begin{aligned}
\log(\det(\mathbf{V})) &= \log(\det(\mathbf{U})\det(\tilde{\mathbf{D}})\det(\mathbf{U}^T)) \\
&= \log\left\{\prod_{i=1}^{N_T}(1 + \eta(\Lambda_i - 1))\right\} \\
&= \sum_{i=1}^{N_T}\log(1 + \eta(\Lambda_i - 1)),
\end{aligned}
\tag{8}
$$

since $\det(\mathbf{U}) = 1$. It also follows from Eq 4 that

$$
\begin{aligned}
\mathbf{V}^{-1} &= (\mathbf{U}\tilde{\mathbf{D}}\mathbf{U}^T)^{-1} \\
&= (\mathbf{U}^T)^{-1}(\tilde{\mathbf{D}})^{-1}\mathbf{U}^{-1} \\
&= \mathbf{U}\tilde{\mathbf{D}}^{-1}\mathbf{U}^T,
\end{aligned}
\tag{9}
$$

since for an orthonormal matrix $\mathbf{U}^{-1} = \mathbf{U}^T$. Substituting Eqs 7, 8 and 9 into Eq 3 the negative

log-likelihood becomes

$$
\begin{aligned}
-\ell(\boldsymbol{\Theta}) \quad &\propto \frac{N_T}{2}\log(\sigma^2) + \frac{1}{2}\sum_{i=1}^{N_T}\log(1 + \eta(\Lambda_i - 1)) + \frac{1}{2\sigma^2}(\mathbf{Y} - \mathbf{X}\boldsymbol{\beta})^T\mathbf{U}\tilde{\mathbf{D}}^{-1}\mathbf{U}^T(\mathbf{Y} - \mathbf{X}\boldsymbol{\beta}) \\
&= \frac{N_T}{2}\log(\sigma^2) + \frac{1}{2}\sum_{i=1}^{N_T}\log(1 + \eta(\Lambda_i - 1)) + \frac{1}{2\sigma^2}(\mathbf{U}^T\mathbf{Y} - \mathbf{U}^T\mathbf{X}\boldsymbol{\beta})^T\tilde{\mathbf{D}}^{-1}(\mathbf{U}^T\mathbf{Y} - \mathbf{U}^T\mathbf{X}\boldsymbol{\beta}) \\
&= \frac{N_T}{2}\log(\sigma^2) + \frac{1}{2}\sum_{i=1}^{N_T}\log(1 + \eta(\Lambda_i - 1)) + \frac{1}{2\sigma^2}(\tilde{\mathbf{Y}} - \tilde{\mathbf{X}}\boldsymbol{\beta})^T\tilde{\mathbf{D}}^{-1}(\tilde{\mathbf{Y}} - \tilde{\mathbf{X}}\boldsymbol{\beta}) \\
&= \frac{N_T}{2}\log(\sigma^2) + \frac{1}{2}\sum_{i=1}^{N_T}\log(1 + \eta(\Lambda_i - 1)) + \frac{1}{2\sigma^2}\sum_{i=1}^{N_T}\frac{\left(\tilde{Y}_i - \sum_{j=0}^{p}\tilde{X}_{ij+1}\boldsymbol{\beta}_j\right)^2}{1 + \eta(\Lambda_i - 1)},
\end{aligned}
$$

(10)

where $\tilde{\mathbf{Y}} = \mathbf{U}^T\mathbf{Y}$, $\tilde{\mathbf{X}} = \mathbf{U}^T\mathbf{X}$, $\tilde{Y}_i$ denotes the $i^{\text{th}}$ element of $\tilde{\mathbf{Y}}$, $\tilde{X}_{ij}$ is the $i, j^{\text{th}}$ entry of $\tilde{\mathbf{X}}$ and $\mathbf{1}$ is a column vector of $N_T$ ones.

## Penalized maximum likelihood estimator

We define the $p + 3$ length vector of parameters $\boldsymbol{\Theta} := (\Theta_0, \Theta_1, \ldots, \Theta_{p+1}, \Theta_{p+2}, \Theta_{p+3}) = (\boldsymbol{\beta}, \eta, \sigma^2)$ where $\boldsymbol{\beta} \in \mathbb{R}^{p+1}, \eta \in [0, 1], \sigma^2 > 0$. In what follows, $p + 2$ and $p + 3$ are the indices in $\boldsymbol{\Theta}$ for $\eta$ and $\sigma^2$, respectively. In light of our goals to select variables associated with the response in high-dimensional data, we propose to place a constraint on the magnitude of the regression coefficients. This can be achieved by adding a penalty term to the likelihood function Eq 10. The penalty term is a necessary constraint because in our applications, the sample size is much smaller than the number of predictors. We define the following objective function:

$$
Q_\lambda(\boldsymbol{\Theta}) = f(\boldsymbol{\Theta}) + \lambda\sum_{j\neq 0}\nu_j P_j(\beta_j),
$$

where $f(\boldsymbol{\Theta}) := -\ell(\boldsymbol{\Theta})$ is defined in Eq 10, $P_j(\cdot)$ is a penalty term on the fixed regression coefficients $\beta_1, \ldots, \beta_{p+1}$ (we do not penalize the intercept) controlled by the nonnegative regularization parameter $\lambda$, and $\nu_j$ is the penalty factor for $j$th covariate. These penalty factors serve as a way of allowing parameters to be penalized differently. Note that we do not penalize $\eta$ or $\sigma^2$. An estimate of the regression parameters $\hat{\boldsymbol{\Theta}}_\lambda$ is obtained by

$$
\hat{\boldsymbol{\Theta}}_\lambda = \arg\min_{\boldsymbol{\Theta}} Q_\lambda(\boldsymbol{\Theta}).
$$

(11)

This is the general set-up for our model. In the next Section we provide more specific details on how we solve Eq 11. We note here that the main difference between the proposed model, and the `lmmlasso` [57], is that we rotate the response vector $Y$ and the design matrix $X$ by the eigen vectors of the kinship matrix. This results in a diagonal covariance matrix making our method orders of magnitude faster and usable for high-dimensional genetic data. A secondary difference is that we are limiting ourselves to a single unpenalized random effect.

## Computational algorithm

We use a general purpose block coordinate gradient descent algorithm (CGD) [58] to solve Eq 11. At each iteration, we cycle through the coordinates and minimize the objective function with respect to one coordinate only. For continuously differentiable $f(\cdot)$ and convex and block-

separable $P(\cdot)$ (i.e. $P(\boldsymbol{\beta}) = \sum_i P_i(\beta_i)$), Tseng and Yun [58] show that the solution generated by the CGD method is a stationary point of $Q_\lambda(\cdot)$ if the coordinates are updated in a Gauss-Seidel manner i.e. $Q_\lambda(\cdot)$ is minimized with respect to one parameter while holding all others fixed. The CGD algorithm has been successfully applied in fixed effects models (e.g. [20, 59]) and linear mixed models with an $\ell_1$ penalty [57]. In the next section we provide some brief details about Algorithm 1. A more thorough treatment of the algorithm is given in S1 Text.

**Algorithm 1**: Block Coordinate Gradient Descent

```
Set the iteration counter k ← 0, initial values for the parameter vec-
tor Θ⁽⁰⁾ and convergence threshold ε;
for λ ∈ {λ_max, ..., λ_min} do
  repeat
```

$$\boldsymbol{\beta}^{(k+1)} \leftarrow \arg\min_{\boldsymbol{\beta}} Q_\lambda(\boldsymbol{\beta}, \eta^{(k)}, \sigma^{2\ (k)})$$

$$\eta^{(k+1)} \leftarrow \arg\min_{\eta} Q_\lambda(\boldsymbol{\beta}^{(k+1)}, \eta, \sigma^{2\ (k)})$$

$$\sigma^{2\ (k+1)} \leftarrow \arg\min_{\sigma^2} Q_\lambda(\boldsymbol{\beta}^{(k+1)}, \eta^{(k+1)}, \sigma^2)$$

```
    k ← k + 1
  until convergence criterion is satisfied: ‖Θ⁽ᵏ⁺¹⁾ - Θ⁽ᵏ⁾‖₂ < ε;
end
```

**Updates for the $\boldsymbol{\beta}$ parameter.** Recall that the part of the objective function that depends on $\boldsymbol{\beta}$ has the form

$$Q_\lambda(\boldsymbol{\Theta}) = \frac{1}{2}\sum_{i=1}^{N_T} w_i\left(\tilde{Y}_i - \sum_{j=0}^{p}\tilde{X}_{ij+1}\boldsymbol{\beta}_j\right)^2 + \lambda\sum_{j=1}^{p}v_j|\boldsymbol{\beta}_j|,$$

where

$$w_i := \frac{1}{\sigma^2(1 + \eta(\Lambda_i - 1))}.$$

Conditional on $\eta^{(k)}$ and $\sigma^{2(k)}$, it can be shown that the solution for $\beta_j, j = 1, \ldots, p$ is given by

$$\boldsymbol{\beta}_j^{(k+1)} \leftarrow \frac{\mathcal{S}_\lambda\left(\sum_{i=1}^{N_T} w_i\tilde{X}_{ij}(\tilde{Y}_i - \sum_{\ell\neq j}\tilde{X}_{i\ell}\boldsymbol{\beta}_\ell^{(k)})\right)}{\sum_{i=1}^{N_T} w_i\tilde{X}_{ij}^2},$$

where $\mathcal{S}_\lambda(x)$ is the soft-thresholding operator

$$\mathcal{S}_\lambda(x) = \mathrm{sign}(x)(|x| - \lambda)_+,$$

$\mathrm{sign}(x)$ is the signum function

$$\mathrm{sign}(x) = \begin{cases} -1 & x < 0 \\ 0 & x = 0 \\ 1 & x > 0 \end{cases},$$

and $(x)_+ = \max(x, 0)$. We provide the full derivation in S1 Text.

**Updates for the $\eta$ paramter.** Given $\boldsymbol{\beta}^{(k+1)}$ and $\sigma^{2(k)}$, solving for $\eta^{(k+1)}$ becomes a univariate optimization problem:

$$\eta^{(k+1)} \leftarrow \arg\min_{\eta} \frac{1}{2} \sum_{i=1}^{N_T} \log(1 + \eta(\Lambda_i - 1)) + \frac{1}{2\sigma^{2\,(k)}} \sum_{i=1}^{N_T} \frac{(\tilde{Y}_i - \sum_{j=0}^{p} \tilde{X}_{ij+1}\boldsymbol{\beta}_j^{(k+1)})^2}{1 + \eta(\Lambda_i - 1)}.$$

We use a bound constrained optimization algorithm [60] implemented in the `optim` function in R and set the lower and upper bounds to be 0.01 and 0.99, respectively.

**Updates for the $\sigma^2$ parameter.** Conditional on $\boldsymbol{\beta}^{(k+1)}$ and $\eta^{(k+1)}$, $\sigma^{2(k+1)}$ can be solved for using the following equation:

$$\sigma^{2\,(k+1)} \leftarrow \arg\min_{\sigma^2} \frac{N_T}{2} \log(\sigma^2) + \frac{1}{2\sigma^2} \sum_{i=1}^{N_T} \frac{(\tilde{Y}_i - \sum_{j=0}^{p} \tilde{X}_{ij+1}\beta_j)^2}{1 + \eta(\Lambda_i - 1)}. \tag{12}$$

There exists an analytic solution for Eq 12 given by:

$$\sigma^{2\,(k+1)} \leftarrow \frac{1}{N_T} \sum_{i=1}^{N_T} \frac{(\tilde{Y}_i - \sum_{j=0}^{p} \tilde{X}_{ij+1}\beta_j^{(k+1)})^2}{1 + \eta^{(k+1)}(\Lambda_i - 1)}.$$

**Regularization path.** In this section we describe how to determine the sequence of tuning parameters $\lambda$ at which to fit the model. Recall that our objective function has the form

$$Q_\lambda(\boldsymbol{\Theta}) = \frac{N_T}{2} \log(\sigma^2) + \frac{1}{2} \sum_{i=1}^{N_T} \log(1 + \eta(\Lambda_i - 1)) + \frac{1}{2} \sum_{i=1}^{N_T} w_i \left( \tilde{Y}_i - \sum_{j=0}^{p} \tilde{X}_{ij+1}\boldsymbol{\beta}_j \right)^2 + \lambda \sum_{j=1}^{p} v_j |\beta_j|. \tag{13}$$

The Karush-Kuhn-Tucker (KKT) optimality conditions for Eq 13 are given by:

$$\begin{aligned}
\frac{\partial}{\partial \beta_1, \ldots, \beta_p} Q_\lambda(\boldsymbol{\Theta}) &= \mathbf{0}_p \\[1em]
\frac{\partial}{\partial \beta_0} Q_\lambda(\boldsymbol{\Theta}) &= 0 \\[1em]
\frac{\partial}{\partial \eta} Q_\lambda(\boldsymbol{\Theta}) &= 0 \\[1em]
\frac{\partial}{\partial \sigma^2} Q_\lambda(\boldsymbol{\Theta}) &= 0.
\end{aligned} \tag{14}$$

The equations in Eq 14 are equivalent to

$$\sum_{i=1}^{N_T} w_i \tilde{X}_{i1}\left(\tilde{Y}_i - \sum_{j=0}^{p}\tilde{X}_{ij+1}\beta_j\right) = 0$$

$$\frac{1}{\nu_j}\sum_{i=1}^{N_T} w_i \tilde{X}_{ij}\left(\tilde{Y}_i - \sum_{j=0}^{p}\tilde{X}_{ij+1}\beta_j\right) = \lambda\gamma_j$$

$$\gamma_j \in \begin{cases} \text{sign}(\hat{\beta}_j) & \text{if} \quad \hat{\beta}_j \neq 0 \\[2mm] [-1,1] & \text{if} \quad \hat{\beta}_j = 0 \end{cases}, \qquad \text{for} \qquad j = 1,\ldots,p \qquad (15)$$

$$\frac{1}{2}\sum_{i=1}^{N_T}\frac{\Lambda_i - 1}{1 + \eta(\Lambda_i - 1)}\left(1 - \frac{(\tilde{Y}_i - \sum_{j=0}^{p}\tilde{X}_{ij+1}\beta_j)^2}{\sigma^2(1 + \eta(\Lambda_i - 1))}\right) = 0$$

$$\sigma^2 - \frac{1}{N_T}\sum_{i=1}^{N_T}\frac{(\tilde{Y}_i - \sum_{j=0}^{p}\tilde{X}_{ij+1}\beta_j)^2}{1 + \eta(\Lambda_i - 1)} = 0,$$

where $w_i$ is given by Eq, $\tilde{\mathbf{X}}_{-1}^T$ is $\tilde{\mathbf{X}}^T$ with the first column removed, $\tilde{\mathbf{X}}_1^T$ is the first column of $\tilde{\mathbf{X}}^T$, and $\gamma \in \mathbb{R}^p$ is the subgradient function of the $\ell_1$ norm evaluated at $(\hat{\beta}_1,\ldots,\hat{\beta}_p)$. Therefore $\hat{\boldsymbol{\Theta}}$ is a solution in Eq 11 if and only if $\hat{\boldsymbol{\Theta}}$ satisfies Eq 15 for some $\gamma$. We can determine a decreasing sequence of tuning parameters by starting at a maximal value for $\lambda = \lambda_{max}$ for which $\hat{\beta}_j = 0$ for $j = 1,\ldots,p$. In this case, the KKT conditions in Eq 15 are equivalent to

$$\frac{1}{\nu_j}\sum_{i=1}^{N_T}|w_i\tilde{X}_{ij}(\tilde{Y}_i - \tilde{X}_{i1}\beta_0)| \leq \lambda, \quad \forall j = 1,\ldots,p$$

$$\beta_0 = \frac{\sum_{i=1}^{N_T} w_i\tilde{X}_{i1}\tilde{Y}_i}{\sum_{i=1}^{N_T} w_i\tilde{X}_{i1}^2}$$

$$\frac{1}{2}\sum_{i=1}^{N_T}\frac{\Lambda_i - 1}{1 + \eta(\Lambda_i - 1)}\left(1 - \frac{(\tilde{Y}_i - \tilde{X}_{i1}\beta_0)^2}{\sigma^2(1 + \eta(\Lambda_i - 1))}\right) = 0 \qquad (16)$$

$$\sigma^2 = \frac{1}{N_T}\sum_{i=1}^{N_T}\frac{(\tilde{Y}_i - \tilde{X}_{i1}\beta_0)^2}{1 + \eta(\Lambda_i - 1)}.$$

We can solve the KKT system of equations in Eq 16 (with a numerical solution for $\eta$) in order to have an explicit form of the stationary point $\hat{\boldsymbol{\Theta}}_0 = \{\hat{\beta}_0, \mathbf{0}_p, \hat{\eta}, \hat{\sigma}^2\}$. Once we have $\hat{\boldsymbol{\Theta}}_0$, we can solve for the smallest value of $\lambda$ such that the entire vector $(\hat{\beta}_1,\ldots,\hat{\beta}_p)$ is 0:

$$\lambda_{max} = \max_j\left\{\left|\frac{1}{\nu_j}\sum_{i=1}^{N_T}\hat{w}_i\tilde{X}_{ij}\left(\tilde{Y}_i - \tilde{X}_{i1}\hat{\beta}_0\right)\right|\right\}, \quad j = 1,\ldots,p.$$

Following Friedman et al. [20], we choose $\tau\lambda_{max}$ to be the smallest value of tuning parameters $\lambda_{min}$, and construct a sequence of $K$ values decreasing from $\lambda_{max}$ to $\lambda_{min}$ on the log scale. The defaults are set to $K = 100$, $\tau = 0.01$ if $n < p$ and $\tau = 0.001$ if $n \geq p$.

**Warm starts.** The way in which we have derived the sequence of tuning parameters using the KKT conditions, allows us to implement warm starts. That is, the solution $\hat{\boldsymbol{\Theta}}$ for $\lambda_k$ is used

as the initial value $\Theta^{(0)}$ for $\lambda_{k+1}$. This strategy leads to computational speedups and has been implemented in the ggmix R package.

**Prediction of the random effects.**   We use an empirical Bayes approach (e.g. [61]) to predict the random effects $\boldsymbol{b}$. Let the maximum a posteriori (MAP) estimate be defined as

$$\hat{\boldsymbol{b}} = \arg \max_{\boldsymbol{b}} f(\boldsymbol{b}|\mathbf{Y}, \boldsymbol{\beta}, \eta, \sigma^2), \tag{17}$$

where, by using Bayes rule, $f(\boldsymbol{b}|\mathbf{Y}, \boldsymbol{\beta}, \eta, \sigma^2)$ can be expressed as

$$
\begin{aligned}
f(\boldsymbol{b}|\mathbf{Y}, \boldsymbol{\beta}, \eta, \sigma^2) \quad &= \frac{f(\mathbf{Y}|\boldsymbol{b}, \boldsymbol{\beta}, \eta, \sigma^2)\pi(\boldsymbol{b}|\eta, \sigma^2)}{f(\mathbf{Y}|\boldsymbol{\beta}, \eta, \sigma^2)} \\
&\propto f(\mathbf{Y}|\boldsymbol{b}, \boldsymbol{\beta}, \eta, \sigma^2)\pi(\boldsymbol{b}|\eta, \sigma^2) \\
&\propto \exp\left\{ -\frac{1}{2\sigma^2}(\mathbf{Y} - \mathbf{X}\boldsymbol{\beta} - \boldsymbol{b})^T(\mathbf{Y} - \mathbf{X}\boldsymbol{\beta} - \boldsymbol{b}) - \frac{1}{2\eta\sigma^2}\boldsymbol{b}^T\boldsymbol{\Phi}^{-1}\boldsymbol{b} \right\} \\
&= \exp\left\{ -\frac{1}{2\sigma^2}\left[ (\mathbf{Y} - \mathbf{X}\boldsymbol{\beta} - \boldsymbol{b})^T(\mathbf{Y} - \mathbf{X}\boldsymbol{\beta} - \boldsymbol{b}) + \frac{1}{\eta}\boldsymbol{b}^T\boldsymbol{\Phi}^{-1}\boldsymbol{b} \right] \right\}.
\end{aligned}
\tag{18}
$$

Solving for Eq 17 is equivalent to minimizing the exponent in Eq 18:

$$\hat{\boldsymbol{b}} = \arg \min_{\boldsymbol{b}} \left\{ (\mathbf{Y} - \mathbf{X}\boldsymbol{\beta} - \boldsymbol{b})^T(\mathbf{Y} - \mathbf{X}\boldsymbol{\beta} - \boldsymbol{b}) + \frac{1}{\eta}\boldsymbol{b}^T\boldsymbol{\Phi}^{-1}\boldsymbol{b} \right\}. \tag{19}$$

Taking the derivative of Eq 19 with respect to $\boldsymbol{b}$ and setting it to 0 we get:

$$
\begin{aligned}
0 \quad &= -2(\mathbf{Y} - \mathbf{X}\boldsymbol{\beta} - \boldsymbol{b}) + \frac{2}{\eta}\boldsymbol{\Phi}^{-1}\boldsymbol{b} \\
&= -(\mathbf{Y} - \mathbf{X}\boldsymbol{\beta}) + \boldsymbol{b} + \left( \frac{1}{\eta}\boldsymbol{\Phi}^{-1} \right)\boldsymbol{b} \\
(\mathbf{Y} - \mathbf{X}\boldsymbol{\beta}) \quad &= \left( \mathbf{I}_{N_T \times N_T} + \frac{1}{\eta}\boldsymbol{\Phi}^{-1} \right)\boldsymbol{b} \\
\hat{\boldsymbol{b}} \quad &= \left( \mathbf{I}_{N_T \times N_T} + \frac{1}{\hat{\eta}}\boldsymbol{\Phi}^{-1} \right)^{-1}(\mathbf{Y} - \mathbf{X}\hat{\boldsymbol{\beta}}) \\
&= \left( \mathbf{I}_{N_T \times N_T} + \frac{1}{\hat{\eta}}\mathbf{U}\mathbf{D}^{-1}\mathbf{U}^T \right)^{-1}(\mathbf{Y} - \mathbf{X}\hat{\boldsymbol{\beta}}),
\end{aligned}
$$

where $(\hat{\boldsymbol{\beta}}, \hat{\eta})$ are the estimates obtained from Algorithm 1.

**Phenotype prediction.**   Here we describe the method used for predicting the unobserved phenotype $\mathbf{Y}^\star$ in a set of individuals with predictor set $\mathbf{X}^\star$ that were not used in the model training e.g. a testing set. Let $q$ denote the number of observations in the testing set and $N - q$ the number of observations in the training set. We assume that a ggmix model has been fit on a set of training individuals with observed phenotype $\mathbf{Y}$ and predictor set $\mathbf{X}$. We further assume that $\mathbf{Y}$ and $\mathbf{Y}^\star$ are jointly multivariate Normal:

$$
\begin{bmatrix} \mathbf{Y}^\star \\ \mathbf{Y} \end{bmatrix} \sim N\left( \begin{bmatrix} \boldsymbol{\mu}_{1_{(q \times 1)}} \\ \boldsymbol{\mu}_{2_{(N-q) \times 1}} \end{bmatrix}, \begin{bmatrix} \boldsymbol{\Sigma}_{11_{(q \times q)}} & \boldsymbol{\Sigma}_{12_{q \times (N-q)}} \\ \boldsymbol{\Sigma}_{21_{(N-q) \times q}} & \boldsymbol{\Sigma}_{22_{(N-q) \times (N-q)}} \end{bmatrix} \right).
$$

Then, from standard multivariate Normal theory, the conditional distribution $\mathbf{Y}^\star | \mathbf{Y}, \eta, \sigma^2, \beta, \mathbf{X}, \mathbf{X}^\star$ is $\mathcal{N}(\mu^\star, \Sigma^\star)$ where

$$\begin{aligned}
\boldsymbol{\mu}^\star &= \boldsymbol{\mu}_1 + \boldsymbol{\Sigma}_{12}\boldsymbol{\Sigma}_{22}^{-1}(\mathbf{Y} - \mu_2) \\
\boldsymbol{\Sigma}^\star &= \boldsymbol{\Sigma}_{11} - \boldsymbol{\Sigma}_{12}\boldsymbol{\Sigma}_{22}^{-1}\boldsymbol{\Sigma}_{21}.
\end{aligned}$$

The phenotype prediction is thus given by:

$$\begin{aligned}
\boldsymbol{\mu}^\star_{q\times 1} &= \mathbf{X}^\star\boldsymbol{\beta} + \frac{1}{\sigma^2}\boldsymbol{\Sigma}_{12}\mathbf{V}^{-1}(\mathbf{Y} - \mathbf{X}\boldsymbol{\beta}) \\
&= \mathbf{X}^\star\boldsymbol{\beta} + \frac{1}{\sigma^2}\boldsymbol{\Sigma}_{12}\mathbf{U}\tilde{\mathbf{D}}^{-1}\mathbf{U}^T(\mathbf{Y} - \mathbf{X}\boldsymbol{\beta}) \\
&= \mathbf{X}^\star\boldsymbol{\beta} + \frac{1}{\sigma^2}\boldsymbol{\Sigma}_{12}\mathbf{U}\tilde{\mathbf{D}}^{-1}(\tilde{\mathbf{Y}} - \tilde{\mathbf{X}}\boldsymbol{\beta}) \\
&= \mathbf{X}^\star\boldsymbol{\beta} + \frac{1}{\sigma^2}\eta\sigma^2\boldsymbol{\Phi}^\star\mathbf{U}\tilde{\mathbf{D}}^{-1}(\tilde{\mathbf{Y}} - \tilde{\mathbf{X}}\boldsymbol{\beta}) \\
&= \mathbf{X}^\star\boldsymbol{\beta} + \eta\boldsymbol{\Phi}^\star\mathbf{U}\tilde{\mathbf{D}}^{-1}(\tilde{\mathbf{Y}} - \tilde{\mathbf{X}}\boldsymbol{\beta}),
\end{aligned}$$

where $\boldsymbol{\Phi}^\star$ is the $q \times (N - q)$ covariance matrix between the testing and training individuals.

**Choice of the optimal tuning parameter.** In order to choose the optimal value of the tuning parameter $\lambda$, we use the generalized information criterion [62] (GIC):

$$GIC_\lambda = -2\ell(\hat{\boldsymbol{\beta}}, \hat{\sigma}^2, \hat{\eta}) + a_n \cdot \hat{df}_\lambda,$$

where $\hat{df}_\lambda$ is the number of non-zero elements in $\hat{\boldsymbol{\beta}}_\lambda$ [63] plus two (representing the variance parameters $\eta$ and $\sigma^2$). Several authors have used this criterion for variable selection in mixed models with $a_n = \log N_T$ [57, 64], which corresponds to the BIC. We instead choose the high-dimensional BIC [65] given by $a_n = \log(\log(N_T)) * \log(p)$. This is the default choice in our `ggmix` R package, though the interface is flexible to allow the user to select their choice of $a_n$.

## Software availability

The `ggmix` method is written in an R package, which is freely available on CRAN at https://cran.r-project.org/package=ggmix. The complete documentation for this package is available at https://sahirbhatnagar.com/ggmix/. Scripts for running the analyses and reproducing the tables and figures reported in the manuscript are available in an `RMarkdown` document at https://github.com/sahirbhatnagar/ggmix/blob/master/manuscript/bin/tables-figures.Rmd.

## Supporting information

**S1 Fig. Distribution of SNPs used in UK Biobank analysis.** Distribution of SNPs used in UK Biobank analysis by chromosome and whether or not the SNP was imputed.
(TIF)

**S2 Fig. LD structure among the markers in the GAW20 dataset.** We illustrate the LD structure among the markers in the GAW20 dataset. We show the pairwise $r^2$ for 655 SNPs within a 1Mb-window around the causal SNP rs9661059 (indicated) that we focused on. The dotplot above the heatmap denotes $r^2$ between each SNP and the causal SNP. It is clear that although strong correlation does exist between some SNPs, none of these nearby SNPs is correlated with the causal SNP. The only dot denoting an $r^2 = 1$ represents the causal SNP itself.
(TIF)

**S3 Fig. LD structure among the markers in the mouse dataset.** We illustrate the LD structure among the markers in the mouse dataset. Shown is the pairwise $r^2$ for all microsatellite markers. It is clear that many markers are considerably strongly correlated with each other, as we expected.
(TIF)

**S1 Text. Block coordinate gradient descent algorithm.** We provide a full derivation of the algorithm used to solve the ggmix objective function.
(PDF)

**S2 Text. ggmix Package Showcase.** We introduce the freely available and open source ggmix package in R available on CRAN (https://cran.r-project.org/package=ggmix).
(PDF)

## Acknowledgments

Part of this research has been conducted using the UK Biobank Resource under project number 27449. We appreciate the generosity of UK Biobank volunteers. MF was at the Lady Davis Institute when she undertook this research. We appreciate advice on an earlier version of the manuscript provided by Dr. Simon Gravel, Dr. David Fardo and Dr. Abbas Khalili.

## Author Contributions

**Conceptualization:** Sahir R. Bhatnagar, Yi Yang, Karim Oualkacha, Celia M. T. Greenwood.

**Data curation:** Sahir R. Bhatnagar, Tianyuan Lu, Erwin Schurr, JC Loredo-Osti, Marie Forest.

**Formal analysis:** Sahir R. Bhatnagar, Tianyuan Lu, Marie Forest.

**Methodology:** Sahir R. Bhatnagar, Yi Yang, Karim Oualkacha, Celia M. T. Greenwood.

**Software:** Sahir R. Bhatnagar, Marie Forest.

**Supervision:** Yi Yang, Karim Oualkacha, Celia M. T. Greenwood.

**Visualization:** Sahir R. Bhatnagar, Tianyuan Lu.

**Writing – original draft:** Sahir R. Bhatnagar, Tianyuan Lu, JC Loredo-Osti, Karim Oualkacha, Celia M. T. Greenwood.

**Writing – review & editing:** Sahir R. Bhatnagar, Yi Yang, Tianyuan Lu, Erwin Schurr, Karim Oualkacha, Celia M. T. Greenwood.

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
