## [Decision Letter · Decision Letter 0]

8 Sep 2019

Dear Dr Bhatnagar,

Thank you very much for submitting your Research Article entitled 'Simultaneous SNP selection and adjustment for population structure in high dimensional prediction models' to PLOS Genetics. Your manuscript was fully evaluated at the editorial level and by independent peer reviewers. The reviewers appreciated the attention to an important problem, but raised some substantial concerns about the current manuscript. Based on the reviews, we will not be able to accept this version of the manuscript, but we would be willing to review again a much-revised version. We cannot, of course, promise publication at that time.

If you decide to revise the manuscript for further consideration at PLOS Genetics, please aim to resubmit within the next 60 days, unless it will take extra time to address the concerns of the reviewers, in which case we would appreciate an expected resubmission date by email to plosgenetics@plos.org.

[LINK]

We are sorry that we cannot be more positive about your manuscript at this stage. Please do not hesitate to contact us if you have any concerns or questions.

Yours sincerely,

Heather J Cordell

Associate Editor

PLOS Genetics

Scott Williams

Section Editor: Natural Variation

PLOS Genetics

Reviewer's Responses to Questions

**Comments to the Authors:**

Reviewer #1: The authors demonstrate a method for genome-wide association that combines mixed-model based control of population structure or genetic relatedness with multi-variate regressions. The idea is to increase power for GWAS applications by including multiple markers at once while accounting for structure. Previous tools for this application are somewhat limited, so this method aims to provide a more comprehensive and computationally efficient tools. They include a well-structured R package to implement their method.

This is an important topic, and their software is much faster and easier to use than existing tools. However I have several comments on their presentation, particularly the comparisons with existing methods and some limitations of the simulations:

Major issues:

1. I think that the novelty of the method is not made clear in the paper. The model itself appears virtually identical to that of the glmmlasso R package (Schelldorfer et al 2011) except limited to one random effect – the objective function appears identical, and those authors also derived a block coordinate gradient descent algorithm, though I don’t know if they are identical. The addition here seems mostly in terms of computational efficiency and flexibility (combinations of L1 and L2 penalties vs just L1 for Schellendorfer). Using the SVD to rotate and diagonalize the LMM so that the random effect covariance matrix is diagonal is a very useful addition and makes the software very fast, but this is not really emphasized in the methods. Also, the BSLMM model (Zhou et al 2013 Plos Genetics) is also effectively a variable selection LMM, though it’s Bayesian instead of frequentist and requires MCMC so is slower. But it’s performance could be compared to ggmix and the two-step LASSO presented in this paper.

2. While this method is much faster than glmmlasso (and presumably BSLMM), how does it scale to large numbers of markers? Typical genomics datasets today contain >1e5-1e6 markers. The datasets used here seem to contain 10K-50K. There is some discussion about limitations of scaling to large N, but not large p.

3. The first set of simulations include population structure and admixture. But all 10K markers are simulated in linkage equilibrium. This is an ideal situation for LASSO. But in real data, nearby markers will be partially correlated. How well does this method select the correct marker when there are other correlated markers? What does it do when the true causal variant is not in the data, but imperfectly correlated markers are (the typical GWAS setting)? Does it tend to select the nearest marker, or does it select a set of nearby markers? I can’t tell in the GAW20 and mouse datasets the extent of the LD among markers and whether this is addressed there. Also, the simulation result that including the causal markers in the kinship matrix has little impact is encouraging. But the Yang et al 2014 paper, which discusses the impact of including causal markers in the estimated kinship matrix, suggests that this is only likely when N/M is large (ie as many or more individuals as markers). In the simulations presented here, N/M is <0.1, which is probably small enough that proximal contamination is not an issue. If N/M were ~1, then the impact might be greater. Note: M is the effective number of markers after accounting for LD among markers.

Minor issues

• 97: Wang et al 2011 does not treat the variance components as fixed, but iteratively estimates them along with beta.

• 142: How much of the total variance is accounted for by causal SNPs vs the background in the simulations?

• 355: X_1,…,X_N should each be transposed

• Fig 5: I don’t think that the red line is a p-value threshold. Maybe “significance threshold”? How can this be <100 if there were 200 bootstraps and significance required >50% inclusion?

• I believe Eq (30) is wrong. The term V^{-1} shouldn’t be present in the likelihood if b is included.

Reviewer #2: I have a few comments and suggestions:

Page 2, Line 33: It this true: better sensitivity and specificity? What are you using to define sensitivity?

Page 7, line 143-163. I found the definitions of the different X matrices confusing. I think you can simplify to indexes: eg: causal for list of causal snp indexes, kinship for list of snp index in the kinship matrix, etc. I’d also refer the the snps as covariates and not label them as fixed, and state when the causal set is in the kinship set.

Page 7, line 159: Did you try a larger number of SNPs? In GWAS the number is much larger and PRS methods use much more than 50 independent SNPs for estimation.

Page 10, line 183: if I understand your sparsity estimate correctly, with a causal rate of 1%, setting all B to zero would give a value of 99%?

Page 11, 188: I am curious why you only reported the ‘optimal’ value of the penalty parameter. Is your method outperforming in terms of sparsity because it just does a better job of selecting a sparse model? Your false positive rate is lower but the true positive rate is much lower. If one is searching for the set of true causal variants, they are usually willing to take the tradeoff of better sensitivity for weeding through the false positives. I would prefer to see curves of FP versus TP rates with the values at the optimal tuning parameter marked. In practice I found AIC/BIC somewhat conservative compared to CV or controlling for error rate via phenotype permutation.

Page 11, Line 196: I am sorry if I missed this somewhere, but how was the model tuned for the lasso and twostep in the training data? Did ggmix use the GIC in the materials and methods?

Page 14, Line 243 typo- methods

Page 12: Figure 3. This data might be better summarized in a table that could include the additional data in the supplementary files.

The math is a bit beyond my abilities, but I have previously read a paper that suggests maximizing the log likelihood, −12[ln|V|+(Y−βX)TV−1(Y−BX)] subject to the L1/L2 norm penalty to control for relatedness in penalized regression methods for genetic data (where V is the variance covariance matrix). Is this essentially what you are doing?

It would have been interesting to use the set of related individuals in the UK Biobank on a few traits where PRS works well.

Reviewer #3: The authors present a penalized multi-variate regression model, ggmix, that jointly models multiple genotypes in mixed-model setup that incorporates the kinship or the genetic relationship matrix (GRM). It is an important problem in statistical genetics and I appreciate that the authors developed a comprehensive algorithm that simultaneously incorporates population structure and variable selection problem. The methods are well described and the paper was easy to follow, but I have some concerns in the experiments and evaluation metric. Overall, I think the paper presents an important problem, but the results are not convincing.

First, in simulation results, the ‘correct sparsity’ measure is basically ‘accuracy’ measure in binary classification problem of whether the regression coefficients are zero or non-zero. This ‘accuracy’ measure is often misleading when class distribution is imbalanced. For example, if 99% of coefficients are zero, you can get 99% accuracy by just classifying everything to be zero, which is clearly not a good model. I suggest adapting different measure, such as MCC. I can see that twostep and LASSO both maintains FPR at 0.05, which is why ‘correct sparsity’ is around 0.95. At the same time, we can see that LASSO and twostep achieve higher TPR. Also from a slightly different point of view, in Figure 3(D), comparing TPR at different points on FPR is not a fair comparison. To compare different methods in the context of TPR and FPR, either AUC or TPR at the same rate of FPR should be considered.

Intro is slightly misleading, especially in lines 107-109, because I first thought that ggmix takes out causal (i.e. selected) variables out of the relationship matrix, then I later realized that loss of power due to causal SNPs included in the GRM still happens in ggmix, but you aim to minimize the loss by joint modeling – but it is not clear why this would be the case. Is there any theoretical or simulation-based evidence that joint modeling achieves higher power in such a case?

In the mouse data, the model parameters were optimized so that the two loci are picked up, and then the evaluation metric is based on whether these two loci are picked up, which is circular.

In discussion, it was mentioned that leave-one-chromosome-out approach is possible, but has not been tried. What would be the compelling reason to model all chromosomes together in the proposed problem, especially when the model is still additive and trans-interaction term is not directly modeled?

**Have all data underlying the figures and results presented in the manuscript been provided?**

Reviewer #1: Yes

Reviewer #2: Yes

Reviewer #3: Yes

PLOS authors have the option to publish the peer review history of their article (what does this mean?). If published, this will include your full peer review and any attached files.

Reviewer #1: No

Reviewer #2: No

Reviewer #3: No

---

## [Decision Letter · Decision Letter 1]

11 Feb 2020

Dear Dr Bhatnagar,

Thank you very much for submitting your Research Article entitled 'Simultaneous SNP selection and adjustment for population structure in high dimensional prediction models' to PLOS Genetics. Your manuscript was fully evaluated at the editorial level and by independent peer reviewers. The reviewers appreciated the attention to an important topic but identified some aspects of the manuscript that should be improved.

We therefore ask you to modify the manuscript according to the review recommendations before we can consider your manuscript for acceptance. Your revisions should address the specific points made by each reviewer.

[LINK]

Yours sincerely,

Heather J Cordell

Associate Editor

PLOS Genetics

Scott Williams

Section Editor: Natural Variation

PLOS Genetics

Reviewer's Responses to Questions

**Comments to the Authors:**

Reviewer #1: The authors have significantly revised their manuscript. I think that the inclusions of BSLMM in the comparisons is useful. However, I have a couple additional concerns.

1. Are you sure you’re extracting the model fit from BSLMM correctly? The specifics of use are not described in the methods. It’s a Bayesian model, so gives a probability of inclusion of each SNP. From the prefix.param.txt file, you should use the gamma column to report the number of SNPs that cross a posterior inclusion probability threshold. If you count how many betas are != 0, that will likely be large. But this is not an accurate estimate of the number of markers included in the model. You can also get the posterior on the number of SNPs included from the prefix.gamma.txt file. If you’re counting SNPs that are included, what posterior probability threshold are you using for inclusion? If you’re reporting the % of SNPs included, are you reporting the posterior mean?

2. Also, the authors didn’t apply BSLMM to several of the analyses. There is a write.plink function in the snpStats package that could be used to write GEMMA-compatible input files from R. Also, the PhenotypeSimulator package seems to have a writeStandardOutput function that can write bimbam or Gemma output. Especially if the extraction of estimates of needs to be revised and in fact works more similarly to the other methods, then I would recommend applying it to all analyses (Biobank may be too large).

3. I am still a bit confused about when tuning parameters were set based on TPR / FPR and when based on GIC or CV or other direct methods. In real data, it’s generally not possible to set based on TPR/FPR, but I think most of your comparisons now are done that way. I understand that the goal is to show that your model can work well, and so comparing to the truth is useful. But I think more clarity is needed about when you’re demonstrating the true performance of the model vs when you’re demonstrating how the model would actually be run by a practitioner who did not know any true positive effects going in.

4. 412-414. Is this statement true? Schelldorfer et al 2011 used what they called a "Block Coordinate Gradient Descent method" for their penalized LMM

5. 501: I think that this is underselling your method. The main difference is that your method is orders of magnitude faster than lmmlasso, so it’s actually usable. A secondary difference is that it is limited to only one random effect.

6. 288: how do you get a p-value from ggmix?

Reviewer #3: The authors addressed most of my major concerns in this revision. I have one more comment:

The authors first state they could not run BSLMM on simulated data and mouse data because of data format issues. Converting genotypes to PLINK format should be straightforward and should not be an issue. I understand for simulating large amount of data, it might be practically difficult, but I cannot understand why mouse microsatellite data cannot be converted to PLINK format. I do not think you need additional experiments with BSLMM but please remove the statement that converting the data to PLINK format is not possible.

**Have all data underlying the figures and results presented in the manuscript been provided?**

Reviewer #1: Yes

Reviewer #3: Yes

PLOS authors have the option to publish the peer review history of their article (what does this mean?). If published, this will include your full peer review and any attached files.

Reviewer #1: No

Reviewer #3: No

---

## [Decision Letter · Decision Letter 2]

8 Apr 2020

Dear Dr Bhatnagar,

We are pleased to inform you that your manuscript entitled "Simultaneous SNP selection and adjustment for population structure in high dimensional prediction models" has been editorially accepted for publication in PLOS Genetics. Congratulations!

Yours sincerely,

Heather J Cordell

Associate Editor

PLOS Genetics

Scott Williams

Section Editor: Natural Variation

PLOS Genetics

Comments from the reviewers (if applicable):

Reviewer's Responses to Questions

**Comments to the Authors:**

Reviewer #1: I am satisfied with the changes made by the authors

**Have all data underlying the figures and results presented in the manuscript been provided?**

Reviewer #1: Yes

PLOS authors have the option to publish the peer review history of their article (what does this mean?). If published, this will include your full peer review and any attached files.

Reviewer #1: No

**Data Deposition**

http://datadryad.org/submit?journalID=pgenetics&manu=PGENETICS-D-19-01153R2

**Press Queries**

---

## [Editor Report · Acceptance letter]

24 Apr 2020

PGENETICS-D-19-01153R2 

Simultaneous SNP selection and adjustment for population structure in high dimensional prediction models 

Dear Dr Bhatnagar, 

We are pleased to inform you that your manuscript entitled "Simultaneous SNP selection and adjustment for population structure in high dimensional prediction models" has been formally accepted for publication in PLOS Genetics! Your manuscript is now with our production department and you will be notified of the publication date in due course.

With kind regards,

Matt Lyles

PLOS Genetics

On behalf of:
